# Humans can infer social preferences from decision speed alone

**Sophie Bavard** [ID] [1]*, **Erik Stuchlý**[1], **Arkady Konovalov**[2], **Sebastian Gluth**[1]

**1** Department of Psychology, University of Hamburg, Hamburg, Germany, **2** Centre for Human Brain Health, School of Psychology, University of Birmingham, Birmingham, United Kingdom

* sophie.bavard@gmail.com

**Data Availability Statement:** All data needed to evaluate the conclusions in the paper are present in the paper and/or the Supplementary Materials, and are available from Github repository https://github.com/sophiebavard/beyond-choices (https://doi.

## Abstract

Humans are known to be capable of inferring hidden preferences and beliefs of their conspecifics when observing their decisions. While observational learning based on choices has been explored extensively, the question of how response times (RT) impact our learning of others' social preferences has received little attention. Yet, while observing choices alone can inform us about the direction of preference, they reveal little about the strength of this preference. In contrast, RT provides a continuous measure of strength of preference with faster responses indicating stronger preferences and slower responses signaling hesitation or uncertainty. Here, we outline a preregistered orthogonal design to investigate the involvement of both choices and RT in learning and inferring other's social preferences. Participants observed other people's behavior in a social preferences task (Dictator Game), seeing either their choices, RT, both, or no information. By coupling behavioral analyses with computational modeling, we show that RT is predictive of social preferences and that observers were able to infer those preferences even when receiving only RT information. Based on these findings, we propose a novel observational reinforcement learning model that closely matches participants' inferences in all relevant conditions. In contrast to previous literature suggesting that, from a Bayesian perspective, people should be able to learn equally well from choices and RT, we show that observers' behavior substantially deviates from this prediction. Our study elucidates a hitherto unknown sophistication in human observational learning but also identifies important limitations to this ability.

## Introduction

Each person's unique set of preferences shapes the decisions they make: by closely observing these decisions, one can gain valuable insights into their likes, dislikes, and priorities. Whether and how one can learn and understand the preferences of others from observing their choices has been well documented in the social and reinforcement learning literatures [1–7]. Yet, focusing solely on choices is often not sufficient to determine the strength of a person's preference (i.e., the confidence with which the person has made their choice or how likely they are to make the same choice again). That is, a person would choose option A if they found it twice as valuable as option B, just as they would if they found option A 10 times as valuable. From the

org/10.5281/zenodo.11178632). All custom scripts
have been made available.

**Funding:** SG is supported by the European
Research Council (ERC) under the European
Union's Horizon 2020 research and innovation
program (Grant agreement No. 948545, https://
cordis.europa.eu/project/id/948545). The funders
had no role in study design, data collection and
analysis, decision to publish, or preparation of the
manuscript.

**Competing interests:** The authors have declared
that no competing interests exist.

**Abbreviations:** BO, Bayes-optimal; DDM, drift-
diffusion model; GLMM, generalized linear mixed
model; RL, reinforcement learning; RT, response
time; WFPT, Wiener first-passage time.

observer's perspective, this leads to a many-to-one relationship between strengths of prefer-
ence and choice, making it impossible to narrow down the strength of preference from choices
alone (unless they can extrapolate from observing multiple choices). Fortunately, the decision-
making process offers more than just choices as an output. It also generates response times
(RTs), which have been found to decrease as the strength of preference increases. In other
words, when faced with equally liked options, individuals tend to take more time to make their
decisions. This negative relationship between RT and utility difference has been established in
many value-based domains, including decisions under risk [8–14], intertemporal choices
[13,15–17], food choices [18–27], happiness measurements [28], and social decision-making
[13,16,29–32].

Despite the critical information on strength of preference provided by RT, their impact on
learning about others' preferences has received limited attention compared to the extensive
study of learning from choices. It has recently been proposed that taking RT into account can
be used to predict the choice of future unseen decisions, even when choices alone would fail to
make correct out-of-sample predictions [14,22,29,33–36]. Most of these studies, however, do
not use RT as information for humans to make inferences on someone else's decision-making
process, but rather as a tool to improve model fitting or model simulation in predicting future
choices. On the other side, recent literature suggests that human adults [24,31,37–39] and chil-
dren [40] do take RT into account when estimating someone else's hidden preference or com-
petence, in paradigms where observers were informed both about the decision-maker's
choices and RT. Importantly though, all these studies only use RT as a supplementary measure
to choices, not as the sole piece of information available to the observer.

On theoretical grounds, this notion was taken even further and it has been proposed that
RT alone (i.e., without observed choices) could be used to infer preferences and predict future
choices. For example, Chabris and colleagues argued that RTs reveal key attributes of the cog-
nitive processes that implement preferences in an intertemporal choice setting [15]. Konovalov
and Krajbich used RT to infer an indifference point in risky choices, social decision-making,
and intertemporal settings [13]. Schotter and Trevino showed that the most informative trial-
based RT has out-of-sample predictive power for determining someone's decision threshold in
a social decision-making setting [29]. So, in principle, it should be possible to infer latent infor-
mation or processes from RT alone, including preferences in value-based decisions. Yet, none
of these studies have tested empirically whether individuals are capable of using RT informa-
tion as effectively and to learn someone else's preference by observing their RT alone.

To answer this question, we propose a preregistered orthogonal design to investigate the
role of both choices and RT in learning and inferring others' social preferences. In our lab
study, participants ($n$ = 46, here referred to as observers) observed other people's decision pro-
cess in a Dictator Game [41,42], where the decision makers ($N$ = 16, here referred to as dicta-
tors) were asked to choose between different monetary allocations between themselves and
another person. Based on their behavior in the Dictator Game, participants can be ranked on a
scale from selfish (choosing the allocation with the higher number of points for themselves) to
prosocial (choosing the allocation with the lower number of points for themselves). Conse-
quently, we assume that the dictators' position on this scale reflects their preferred allocation:
their ideal ratio of points for themselves versus the other person. Therefore, a decision problem
with 2 options equally distant from the preferred allocation represents a choice between 2
equally liked allocations, resulting in high decision difficulty and the expectation that RT
should be very long. Conversely, if the options' distances to the preferred allocation are
unequal (in other words, one option is much closer to the preference), this results in low deci-
sion difficulty and the expectation that RT should be very short. In this framework, we varied
the amount of information provided to the observers: choice and RT information was either

hidden or revealed to observers in a 2-by-2 within-subject design. Behavioral analyses confirmed our hypothesis, as observers were able to learn the dictators' social preferences when they could observe their choices, but also when they could only observe their RT. To gain mechanistic insights into these observational learning processes, we developed a reinforcement learning (RL) model that takes both choices and RT into account to infer the dictator's social preference. This model closely captured the performance and learning curves of observers in the different conditions. On the other side, recent studies have proposed (inverted) Bayesian inference as the optimal framework underlying the cognitive process of social learning [24,43–47], and (quasi-)optimal Bayesian learning has been reported in various fields such as reward-based learning [48,49] or multisensory integration [50]. Motivated by this work, we designed a benchmark Bayes-optimal (BO) model in which the observer's belief on the dictator's social preferences and choice processes is updated using Bayes' rule on prior and current observations. By comparing this BO model to the RL model, we show that, while observers' learning is close to optimal when they can observe choices, they substantially deviate from optimality when they can only observe RT, suggesting that the underlying mechanisms are better captured by our approximate reinforcement learning model. Overall, our study proposes an innovative approach to investigate the role of RT in learning and inferring preferences, identifies a new sophistication in human social inferences, and highlights the importance of considering a greater extent of decision processes when investigating observational learning.

## Results

### Experimental protocol

To test whether people learn someone else's social preference when observing only their RTs, we designed a two-task experiment involving a variant of the Dictator Game [41,42]. In this variant, participants were asked to choose between 2 two-color circles, each representing a proportion of points allocated to themselves ("self") and to another person ("other," **Fig 1A**). For the "Dictator task," we recruited a sample of 16 participants, which will be referred to as dictators, and we recorded both their choices and RT. For the "Observer task," we recruited a sample of 46 participants. These participants, which will be referred to as observers, were asked to first complete a shortened version of the Dictator task, before observing the (previously recorded) decisions of all 16 dictators. For the observation phase, we used a $2 \times 2$ within-subject orthogonal design, manipulating the amount of information provided to the observers: the dictator's choices revealed or hidden, their RT revealed or hidden (**Fig 1B**). Before observing the decisions of a dictator, observers were informed that they were about to observe a new person's decisions, and whether they would see their choices, RT, both, or no information. They were asked to estimate the social preference of this person (their most preferred allocation), once before observing any decision and then after every 4 trials, for a total of 4 estimations over the 12 observed trials per dictator (estimation trials, **Fig 1C**). After observing the 12 trials, observers were asked to predict what this person would choose in 4 previously unseen decision problems (prediction trials, **Fig 1C**). After these 4 prediction trials, the instruction screen for a new dictator was presented. Crucially, all observed and predicted trials were decision problems that observers completed for themselves in the Dictator Game task before observing the dictators.

### Dictator Game results

We first ascertained that the social preference could, in principle, be learned in all conditions, i.e., that the dictator's choices and RT would be good predictors of their social preference. In our task, their social preference refers to the same construct as their preferred allocation,

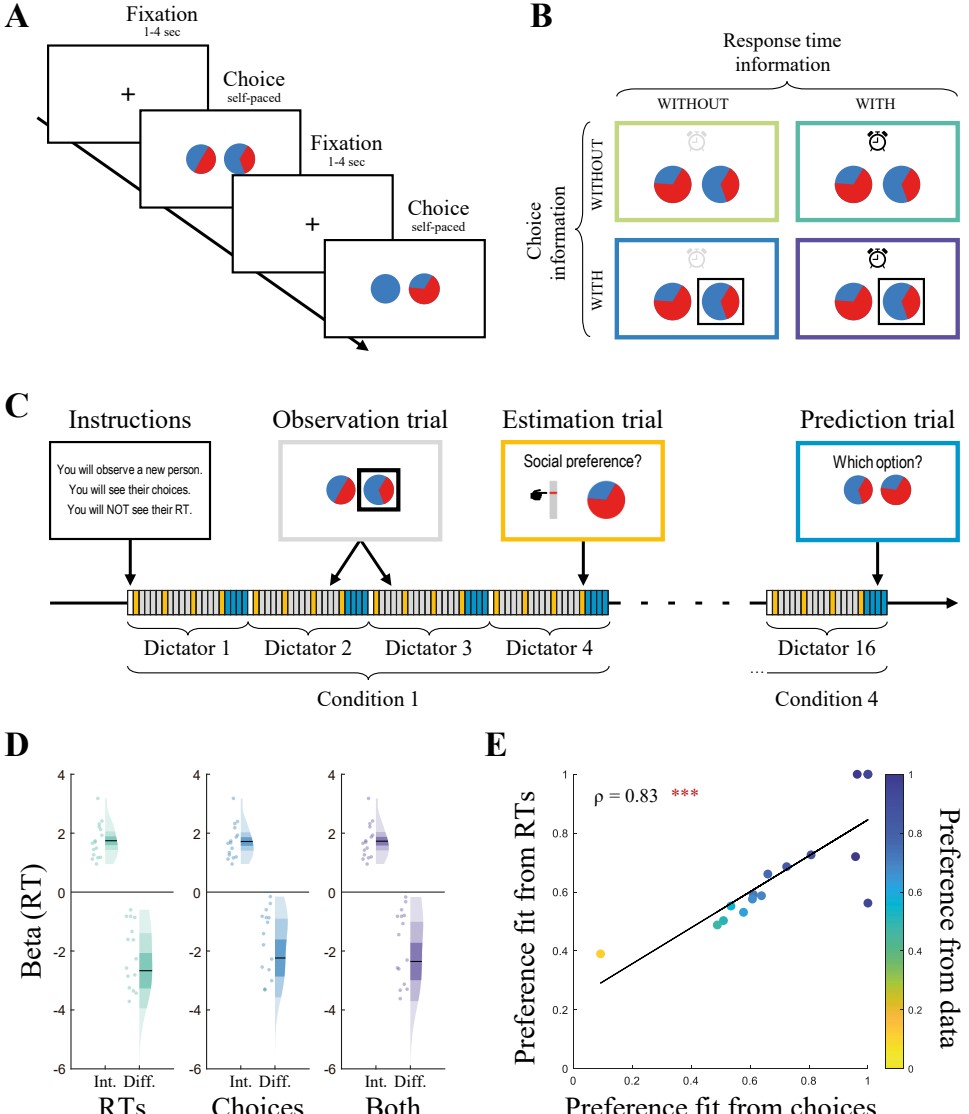

**Fig 1. Task design and validation of the social preference measure.** **(A)** Trial sequence of the Dictator Game part. **(B)** Orthogonal design of the observation part. Observers were presented with 4 successive conditions varying in the visibility of the choice information (with or without, represented with a black square around the chosen option) and the RT information (with or without, represented with a time interval between allocations onset and choice onset). Both allocations were displayed in all conditions. **(C)** Task design of the observation part. Observers were explicitly informed that they were about to observe a new dictator's decisions, and in which condition. After observing all trials of a said dictator, they were asked to predict what this person would choose in previously unseen decision problems. **(D)** Regression coefficients with RT per trial as the dependent variable and trial difficulty as the independent variable, for the complete Dictator Game task performed by the dictators. The difficulty was estimated as the difference in subjective values between both allocations (**Eq 2**), using the preference fitted as a free parameter from RTs only (left), choices only (middle), or both (right). Int.: intercept. Diff.: difficulty. Points indicate individual average, shaded areas indicate probability density function, 95% confidence interval, and SEM. $N = 16$. **(E)** Dictators' social preference fitted from the RTs alone as a function of their preference fitted from the choices alone, and their preference extracted from behavioral data in the Dictator Game task (**Eq 2**). ρ: Spearman's coefficient. $N = 16$. ***$p < 0.001$. Data and analysis scripts underlying this figure are available at https://github.com/sophiebavard/beyond-choices.

defined earlier as their ideal ratio of points for themselves versus the other person. On each trial, we calculated subjective values $s(\cdot)$ for each option (left and right), using the social preference estimated as a free parameter (see **Materials and methods**):

$$s(\text{left}) = 1 - (\text{left} - \text{Pref})^2$$

where left is the objective value of the left (resp. right) option (i.e., the number of points allocated to "self") and Pref is the fitted social preference (see **S1 Text** for more details). Therefore, an option close to the social preference will have a higher subjective value. We regressed the RT with the difference in subjective values between both options and found a negative effect of this difference in all conditions for all 16 dictators, suggesting that decision problems with options of similar subjective values produce longer RT (**Fig 1D**). In addition, we found a significant positive correlation between the preference estimated from the choices only fitting a softmax rule, and the preference estimated from the RT only fitting a DDM (Spearman's $\rho(14)$ = 0.83, $p < 0.0001$, **Fig 1E**), replicating previous results [13]. This suggests that both information types are not only sufficient on their own to make inferences on someone else's social preference, but also lead to inferring the same preference. Together, these results show that, in our Dictator task, RT is a good predictor of social preference as captured by the subjective values (**Eq 2**).

## Observational learning results

After showing that social preference was a good indicator of how long it takes one to make a decision in our task, we turned to the main experiment. The main goal of this study is to investigate whether observers can effectively learn someone else's social preference by observing their decisions, and more specifically either their RT alone, choices alone, or both (**Fig 1D and 1E**). To this end, we selected 12 trials per dictator to be observed by the observers (see **Materials and methods** and **S2 Fig** for more details on the trial selection). To assess learning during the task, observers were asked to estimate the dictator's preference on several occasions: once before any observation, then after each 4 trials. First, in accordance with our preregistered analyses, we found significant correlations between the observers' own preference and (1) their first estimation (before any observation; Spearman's $\rho(44)$ = 0.38, $p = 0.0099$, **S2A Fig**), as well as (2) their average estimation, depending on the amount of information provided to them (average estimation per condition; none: Spearman's $\rho(44)$ = 0.56, $p < 0.0001$; RT only: Spearman's $\rho(44)$ = 0.48, $p = 0.0018$; choice only: Spearman's $\rho(44)$ = 0.31, $p = 0.038$; both: Spearman's $\rho(44)$ = 0.27, $p = 0.073$, **S2B Fig**). Then, according to our main preregistered hypothesis, we analyzed observers' accuracy in estimating the dictators' preference (note that for readability, statistical tests of this paragraph are summarized in **Table 1** rather than reported in the text). On average, observers were able to learn above the empirical chance level (see **Materials and methods**, $t(45)$ = 22.59, $p < 0.0001$, $d$ = 3.33), even in the "RT only"

**Table 1. Pairwise comparisons of average accuracy per condition.** Emp: empirical, ** $p < 0.01$, *** $p < 0.001$, Bonferroni-corrected for between-conditions pairwise comparisons. Data and analysis scripts underlying this table are available at https://github.com/sophiebavard/beyond-choices.

|  | none | | | RT only | | | choice only | | | both | | |
|---|---|---|---|---|---|---|---|---|---|---|---|---|
|  | t-value | p-value | effect size | t-value | p-value | effect size | t-value | p-value | effect size | t-value | p-value | effect size |
| emp. chance level | 6.89 | <0.0001*** | 1.02 | 14.89 | <0.0001*** | 2.20 | 20.04 | <0.0001*** | 2.95 | 23.80 | <0.0001*** | 3.51 |
| none | - | - | - | 3.45 | 0.0073** | 0.64 | 8.37 | <0.0001*** | 1.37 | 8.71 | <0.0001*** | 1.51 |
| RT only | - | - | - | - | - | - | 4.54 | 0.00025** | 0.90 | 5.88 | <0.0001*** | 1.06 |
| choice only | - | - | - | - | - | - | - | - | - | 0.53 | 1.0 | 0.09 |

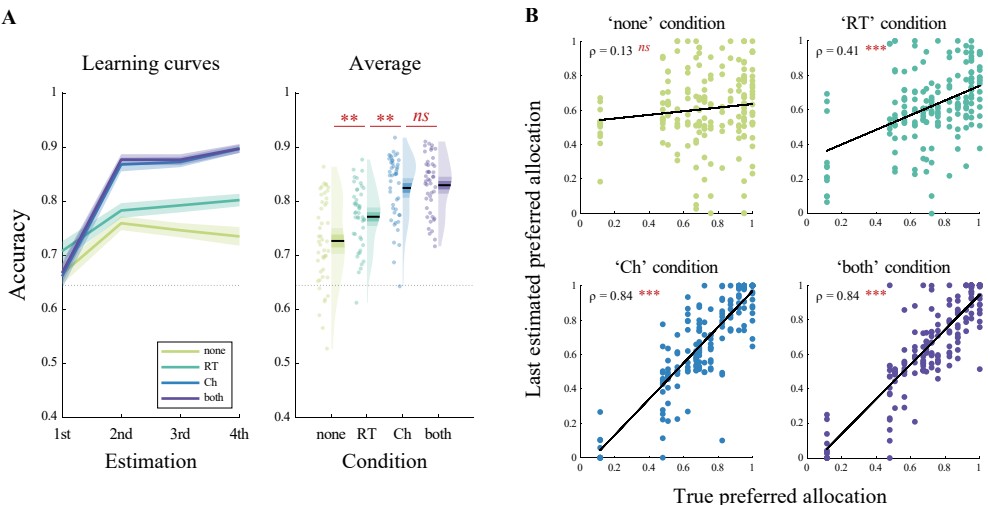

**Fig 2. Behavioral results in the estimation phase.** (A) Observers' accuracy for each estimation as a function of the condition (choice and RT visibility). Left: learning curves; right: average across all trials. Points indicate individual average, shaded areas indicate probability density function, 95% confidence interval, and SEM. $N = 46$. (B) Reported fourth and last estimation per observer per observed dictator, as a function of the true preference of each dictator, for each condition. $N = 184$. ρ: Spearman's coefficient. In all panels, ns: $p > 0.05$, **$p < 0.01$, ***$p < 0.001$, Bonferroni-corrected for pairwise comparisons. Data and analysis scripts underlying this figure are available at https://github.com/sophiebavard/beyond-choices.

condition (**Fig 2A**). Surprisingly, observers' accuracy was above the empirical chance level in the "none" condition as well (**Fig 2A**). However, the correlation between the dictators' true preference and the observers' last estimation was not significant in this condition (Spearman's $\rho(182) = 0.13$, $p = 0.071$), whereas it was significant in all conditions where some information was provided (RT only: Spearman's $\rho(182) = 0.41$, $p < 0.0001$; choice only: Spearman's $\rho(182) = 0.84$, $p < 0.0001$; both: Spearman's $\rho(182) = 0.84$, $p < 0.0001$). This suggests that observers learned to distinguish more prosocial from more selfish dictators in conditions with information but not in the "none" condition, where they mostly used their own preference (**Figs 2B** and **S3**). Furthermore, while accuracy was higher in the "both" condition than in the "RT only" condition, observers seemed to learn equally well in the "choice only" and "both" conditions (**Fig 2A**). The latter result was in contrast with our predictions. All statistical analyses across conditions are reported in **Table 1**. Finally, to get a more fine-grained understanding of learning dynamics, we amended the preregistered analysis and performed a 4 × 4 ANOVA with factors condition ("none," "RT only," "choice only," "both") x estimation number (1st, 2nd, 3rd, 4th). Consistent with our results so far, we found significant main effects of both conditions ($F(3,135) = 36.84$, $p < 0.0001$, $\eta^2 = 0.45$, Huynh–Feldt corrected) and estimation number ($F(3,135) = 80.71$, $p < 0.0001$, $\eta^2 = 0.64$, Huynh–Feldt corrected), and more interestingly a significant interaction ($F(9,405) = 13.58$, $p < 0.0001$, $\eta^2 = 0.23$, Huynh–Feldt corrected), suggesting that observers learned faster in the "choice only" and "both" conditions (**Fig 2A**).

## Extrapolation to unseen decisions

After having observed all 12 trials of a dictator, we asked the observers to predict what the dictator's choices would be in a series of 4 previously unseen trials (see **Materials and methods** for more details on the trial selection). From here on, in contrast with "choice only" and "RT only" conditions, we define "choice visibility" and "RT visibility" as orthogonal factors in our design representing whether or not the choice (resp. RT) information was displayed in each

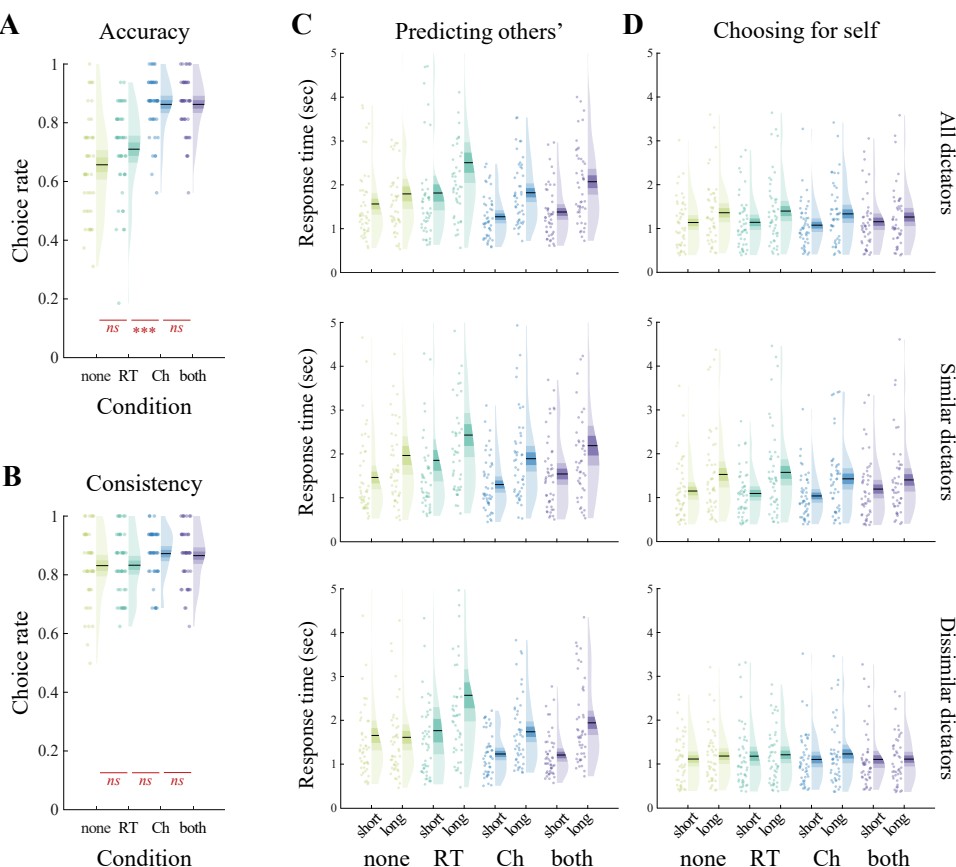

**Fig 3. Behavioral results in the prediction phase.** (A) Observers' accuracy (correct choice rate, i.e., whether they chose the same allocation as the dictator) as a function of the condition (choice and RT visibility). (B) Observers' consistency (choice rate, i.e., whether or not the chosen allocation is consistent with their last estimation for each particular dictator) as a function of the condition (choice and RT visibility). (C) Observers' RT when predicting the dictators' decision, as a function of the dictator's RT for each condition. Top: average for all 16 dictators; middle: average for 8 similar dictators only; bottom: average for 8 dissimilar dictators only. (D) Observers' RT when choosing for themselves, only in trials corresponding to the decisions they had to predict. Top: average over the corresponding trials of all 16 dictators; middle: average over the corresponding trials of the 8 similar dictators; bottom: average over the corresponding trials of the 8 dissimilar dictators. In all panels, points indicate individual average, shaded areas indicate probability density function, 95% confidence interval, and SEM. $N = 46$. ns: $p > 0.05$, ***$p < 0.001$, Bonferroni-corrected for pairwise comparisons. Data and analysis scripts underlying this figure are available at https://github.com/sophiebavard/beyond-choices.

condition; for example, choice visibility is set to 1 in the "choice only" and "both" conditions, and to 0 in the "RT only" and "none" conditions. We first looked at the accuracy, i.e., whether the observer chose the same option as the dictator. In line with the estimation phase results, we found a main effect of choice visibility on prediction accuracy ($F_{(1,45)} = 91.52$, $p < 0.0001$, $\eta_p^2 = 0.67$), but no effect of RT visibility ($F_{(1,45)} = 2.07$, $p = 0.16$, $\eta_p^2 = 0.04$) and no interaction ($F_{(1,45)} = 2.20$, $p = 0.15$, $\eta_p^2 = 0.05$, **Fig 3A**). We then looked at the consistency, i.e., whether the observer's choice was consistent with their last preference estimation of the dictator. We found a small main effect of choice visibility ($F_{(1,45)} = 5.61$, $p = 0.022$, $\eta_p^2 = 0.11$), but no effect of RT visibility ($F_{(1,45)} = 0.041$, $p = 0.84$, $\eta_p^2 = 0.00$) and no interaction ($F_{(1,45)} = 0.12$, $p = 0.73$, $\eta_p^2 = 0.00$, **Fig 3B**). Overall, both the average accuracy and consistency were higher than the chance level of 0.5 (accuracy: $t_{(45)} = 23.11$, $p < 0.0001$, $d = 3.41$; consistency: $t_{(45)} = 36.01$, $p < 0.0001$, $d = 5.31$), suggesting that observers were able to extrapolate their learning of the

dictators' social preference to previously unseen decision problems, and they did so in accordance with their last estimation.

## Observers' prediction speed mimics dictators' decision speed

The analyses of observers' accuracy when predicting decisions confirm that they efficiently learned the dictators' social preferences and were able to use this information to infer which future decisions might be made in previously unseen contexts. Yet, the inspection of their choices alone does not provide much information about the underlying mechanisms and dynamics of how observers predict others' decisions. To dig deeper into these mechanisms, we analyzed observers' RT during the prediction phase. Unbeknownst to the observers, they always predicted 2 easy decisions (where the dictator's RT was fast) and 2 hard decisions (where the dictator's RT was slow, **S1 Fig**). We ran a generalized linear mixed model (GLMM), regressing the observers' RT onto the independent variables: choice visibility (in the estimation phase), RT visibility (in the estimation phase), and trial duration (i.e., whether the dictator's RT was short or long). We found a significant main effect of choice visibility (estimate = −0.26, SE = 0.086, $t$ = −3.04, $p$ = 0.0024, **Table 2**), suggesting that observers made overall faster predictions when the choice information had been available in the estimation phase. The main effect of RT visibility was also significant (estimate = 0.19, SE = 0.091, $t$ = 2.05, $p$ = 0.041, **Table 2**), suggesting observers were overall slower to predict when the RT information had been available in the estimation phase. For interaction effects, please refer to **Table 2**.

Critically, we also found a main effect of trial duration (estimate = 0.32, SE = 0.086, $t$ = 3.71, $p$ = 0.00021, **Table 2**), suggesting that a choice set that elicited a long RT for the dictator also elicited a long RT for the observer (**Fig 3C**, top). This main effect of trial duration is particularly interesting as it suggests that observers put themselves in the shoes of the dictator and predicted the decision in line with the dictator's perceived difficulty. Under this assumption, one would expect observers to show a long RT when predicting decisions that were hard for the dictator, even if the observer themselves found the decision to be easy (and vice versa). Importantly, however, this pattern should only emerge in the 3 conditions, in which observers could learn inter-individual differences in social preferences, that is, in the "both," "choice only," and "RT only" conditions, but not in the "none" condition.

To test this hypothesis, we leveraged the fact that option sets in the "prediction" stage were a subset of option sets in the "self" stage. We then categorized all dictators based on how similar their social preferences were to each of the observers and performed the same regression as

**Table 2. Results from GLMM fitted on observers' RT in the prediction phase.** The GLMM (generalized linear mixed model with Gamma distribution and identity link function) was fitted on the observers' RT, with choice visibility in the estimation phase, RT visibility in the estimation phase, and trial duration (i.e., whether the dictator's RT was short or long), as independent variables. Denotation: Du = duration (fast or slow), Ch = choice visibility (displayed or not), RT = RT visibility (displayed or not), \*\*\*$p$ < 0.001, \*\*$p$ < 0.01, \*$p$ < 0.05. Data and analysis scripts underlying this table are available at https://github.com/sophiebavard/beyond-choices.

| Effect | Predicting others' decision | | | | Similar dictators | | | | Dissimilar dictators | | | |
|---|---|---|---|---|---|---|---|---|---|---|---|---|
| | Estimate | Std. Error | *t*-value | *p*-value | Estimate | Std. Error | *t*-value | *p*-value | Estimate | Std. Error | *t*-value | *p*-value |
| Intercept | 1.54 | 0.11 | 14.54 | <0.0001 \*\*\* | 1.49 | 0.12 | 12.30 | <0.0001 \*\*\* | 1.61 | 0.12 | 13.54 | <0.0001 \*\*\* |
| Duration | 0.32 | 0.086 | 3.71 | 0.00021 \*\*\* | 0.47 | 0.12 | 4.09 | <0.0001 \*\*\* | 0.15 | 0.099 | 1.54 | 0.12 |
| Choice visibility | −0.26 | 0.086 | −3.04 | 0.0024 \*\* | −0.14 | 0.10 | −1.33 | 0.18 | −0.38 | 0.12 | −3.16 | 0.0016 \*\* |
| RT visibility | 0.19 | 0.091 | 2.05 | 0.041 \* | 0.28 | 0.12 | 2.32 | 0.020 \* | 0.098 | 0.10 | 0.93 | 0.35 |
| Du x Ch | 0.27 | 0.082 | 3.25 | 0.0012 \*\* | 0.16 | 0.11 | 1.42 | 0.16 | 0.35 | 0.11 | 3.16 | 0.0016 \*\* |
| Du x RT | 0.37 | 0.090 | 4.01 | <0.0001 \*\*\* | 0.21 | 0.13 | 1.71 | 0.088 | 0.54 | 0.12 | 4.46 | <0.0001 \*\*\* |
| Ch x RT | −0.038 | 0.076 | −0.50 | 0.62 | −0.055 | 0.10 | −0.52 | 0.60 | −0.046 | 0.10 | −0.45 | 0.65 |
| Du x Ch x RT | −0.28 | 0.12 | −2.24 | 0.025 \* | −0.21 | 0.17 | −1.25 | 0.21 | −0.30 | 0.17 | −0.180 | 0.073 |

**Table 3. Results from GLMM fitted on observers' RT in the Dictator Game task.** The GLMM (generalized linear mixed model with Gamma distribution and identity link function) was fitted on the observers' RT when choosing for themselves in their Dictator Game task. Denotation: ***$p < .001$. Data and analysis scripts underlying this table are available at https://github.com/sophiebavard/beyond-choices.

| Effect | Choosing for self | | | | Similar dictators | | | | Dissimilar dictators | | | |
|---|---|---|---|---|---|---|---|---|---|---|---|---|
| | Estimate | Std. Error | *t*-value | *p*-value | Estimate | Std. Error | *t*-value | *p*-value | Estimate | Std. Error | *t*-value | *p*-value |
| Intercept | 1.19 | 0.090 | 13.23 | <0.0001 *** | 1.19 | 0.089 | 13.35 | <0.0001 *** | 1.22 | 0.095 | 12.92 | <0.0001 *** |
| Duration | 0.23 | 0.043 | 5.23 | <0.0001 *** | 0.40 | 0.079 | 5.01 | <0.0001 *** | 0.066 | 0.037 | 1.79 | 0.074 |

reported above on "prediction" RTs for the similar and dissimilar groups of dictators. As expected, we found that observers' prediction RT were longer for long RT of similar as well as dissimilar dictators in the "both," "choice only," and "RT only" conditions (**Fig 3C**, middle and bottom). In the "none" condition, however, this effect was only seen for similar dictators. In line with these patterns, the regression analyses revealed a significant main effect of duration for similar dictators (as the effect was present in all 4 conditions) but significant interactions of duration with both choice and RT visibility for dissimilar dictators (as the effect was not present in the "none" condition") (**Table 2**). These results are consistent with the notion that observers put themselves in the shoes of the dictator whenever they could learn the dictator's individual social preferences. In the "none" condition, however, observers most likely used on their own preferences to make predictions (in line with the findings of the estimation phase; **S2B Fig**). Thus, because of the high match of easy versus difficult choice sets with similar but not dissimilar dictators, the duration effect on prediction RT was seen in the former but not the latter case.

To further substantiate this interpretation, we also applied the regression model to the "self" RT on the trials shared with both types of dictators. Here, we would expect the duration effect (short versus long) to be present for trials shared with the similar dictator in all conditions, but to be entirely absent for trials shared with the dissimilar dictator. Indeed, the duration effect was significant for the trials shared with similar dictators (estimate = 0.40, SE = 0.079, $t = 5.01$, $p < 0.0001$, **Table 3** and **Fig 3D**, middle) but not for those shared with the dissimilar dictators (estimate = 0.066, SE = 0.037, $t = 1.79$, $p = 0.074$, **Table 3** and **Fig 3D**, bottom).

Together, these results all converge to suggest that (1) observers were able to extrapolate the learned social preference to predict decisions for someone else, even if this person had dissimilar social preferences; (2) if a trial was difficult for the dictator, it was also difficult to predict for the observer; (3) whether or not the decision problem was difficult for the observer themselves did not impact how difficult it was for them to predict the dictator. To conclude, in our task, observers were not only able to learn other people's social preference, but they also applied this information to make decisions for this individual that matched the person's preferences and decision dynamics, even though they would behave differently when choosing for themselves.

## Computational formalization of the behavioral results

Behavioral analyses confirmed our hypothesis: trial-by-trial, observers were able to learn the dictators' social preferences when they could observe their choices, but also when they could only observe their RT. To gain a more thorough understanding of the mechanisms underlying social preference learning on the basis of observing different features of the decision process, we developed a modified version of a well-established reinforcement learning model [51,52]. To infer the dictator's social preference, the model takes both choice and RT information (if available) into account, as well as features of the choice options. At each trial *t*, the estimated

preference $P$ is updated with a delta rule:

$$P_t = P_{t-1} + \alpha * \delta_t$$

where $\alpha$ is the learning rate and $\delta_t$ is a prediction error term, calculated as the difference between the outcome $O_t$ (defined below) and the current estimation:

$$\delta_t = O_t - P_{t-1}$$

The outcome $O_t$ depends on the type and the amount of information provided to the observer (RL model, see **Materials and methods**). Intuitively, when only the choice information is available, the outcome is computed as whether or not the chosen option was the more selfish one. When the RT information is available, it is used to categorize the decision between fast and slow. In case of observing a slow decision, the outcome is always computed as the midpoint between the objective values of both options (see **S1 Text** for more details). In case of observing a fast decision, the outcome depends on whether or not the choice information was displayed. If yes, it is computed as whether or not the chosen option was the more selfish one. If not, the observer is assumed to believe that the option with the higher subjective value was chosen. Finally, when no information was displayed, the outcome was computed as the midpoint between the objective values of both options, implying that in case of receiving no information, the observer is assumed to believe that the dictator was asked to make very difficult decisions (and thus decisions that would be diagnostic of their social preference).

The model closely captures several key aspects of observers' behavior. In particular, it matches observers' accuracy in all conditions (**Fig 4A and 4B**), as well as their last

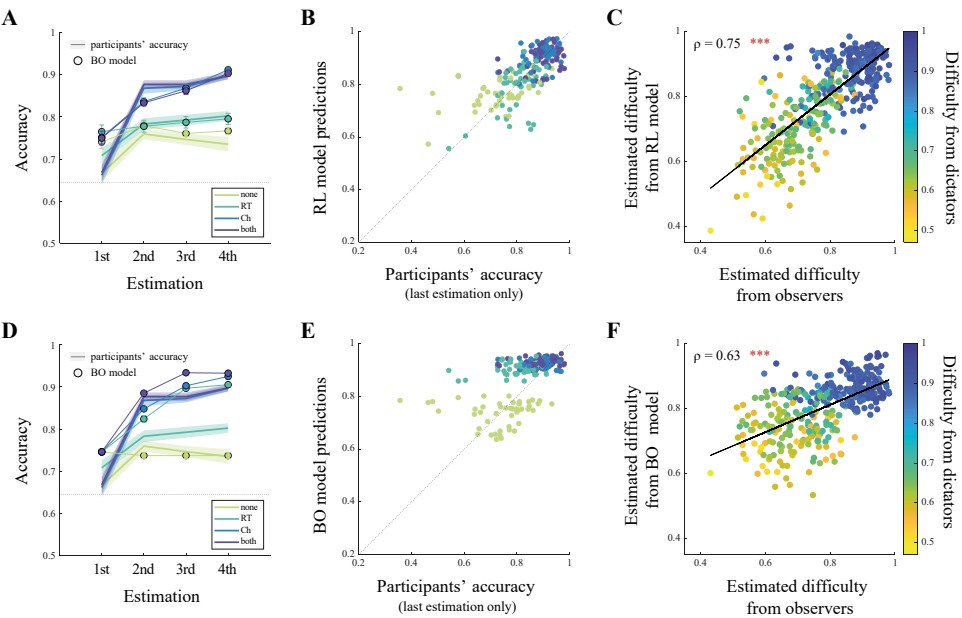

**Fig 4. Qualitative model comparison. (A, D)** Simulated data (colored dots) superimposed on behavioral data (colored curves) representing the accuracy in the estimation phase for the RL model **(A)** and the BO model **(D)** in each condition. Shaded areas represent SEM. $N = 46$. **(B, E)** RL model **(B)** and BO model **(E)** accuracy predictions as a function a behavioral accuracy in the estimation phase for the last estimation of each participant in each condition. Dashed line represents identity. $N = 184$. **(C, F)** Estimated difficulty extracted from RL model **(C)** and BO model **(F)** predictions as a function of the estimated difficulty from behavioral data from observers and dictators, after the estimation phase, for trials from the prediction phase. Each point represents 1 average trial difficulty for each duration (fast/slow) for each condition for each observer. $N = 368$. ***$p < 0.001$. Data and analysis scripts underlying this figure are available at https://github.com/sophiebavard/beyond-choices.

estimation per dictator (**S4C Fig**). Besides matching accuracy, our model was able to reproduce the difficulty patterns (our best proxy for RT, which are not simulated by our model; **Figs 4C** and **S4A**). In addition, the RL model also captured observers' choices in the prediction phase (**S4B Fig**). To compare the model and the empirical data of our study to an optimal benchmark, we designed a Bayes-optimal inference model (BO model) that learns the social preference by updating the posterior probability of the model's parameters, given the available information (see **Materials and methods**). We found that, while observers' learning is close to optimal when they can observe choices, they substantially deviate from optimality when they can only observe RT (BO model predictions versus behavioral last estimation: Spearman's $\rho(44) = 0.10$, $p = 0.52$; RL model predictions versus behavioral last estimation: Spearman's $\rho(44) = 0.47$, $p = 0.0011$; Fisher's $z = 1.89$, $p = 0.029$; **Fig 4D–4F**). Actually, while it is able to match observers' behavior when they predict dictators' decisions, the BO model was unable to match observers' accuracy in all conditions, contrary to the RL model (**S5 Fig**). Together, these modeling results suggest that the computational mechanisms underlying RT-based observational learning are better captured by our approximate RL model.

## Discussion

Humans and other animals are known to learn not only by experiencing rewards and punishments themselves, but also by observing others' actions and outcomes. On the one hand, this allows learning from punishments and losses without incurring these negative outcomes directly, which comes with obvious evolutionary benefits [7]. On the other hand, observing others can reveal information about their beliefs and preferences, which may be critical for future interactions [53]. So far, research on observational learning has focused on testing whether and how people learn from others' choices but has largely ignored other sources of information. Here, we set out to fill this gap by studying the computational mechanisms of learning from observing (only) the speed with which decisions are made. We find that people are, indeed, capable of learning from observing RTs only, but that—contrary to previous assertions [24,31,37]—this ability falls short of an optimal Bayesian learner and is instead better described by an RL model.

In the Dictator Game, where one participant has the power to allocate money to another participant, the RT of the dictator can provide insights into their underlying social preferences. When individuals have a clear and strong preference for a particular allocation, they tend to respond quickly and assertively. However, when faced with a decision where their preferences are less well-defined or when considering 2 options with similar appeal, individuals often exhibit longer RT, indicating hesitation or conflict in their decision-making process. This illustrates how RT can serve as a window into the underlying social preferences of individuals: RT can be used as cues to infer other people's social preferences. Their influence extends beyond the choices individuals make, as RTs are intimately related to the cognitive decision-making processes and reflect the complex interplay between preferences, beliefs, and social context. Building on previous studies, which either suggested RT to be an important source of information theoretically [13,15,29] or showed that humans do use RT to improve their predictions [24,31,37], we designed a task where participants observed someone else's decisions and had to estimate their underlying preference and predict their future decisions. Combining a factorial design that systematically varied the available sources of information, with asking participants to observe, estimate, and predict individual dictators over repeated trials, allowed us to go beyond existing work in characterizing the computational mechanisms of observational learning from different decision process in great detail.

First, we showed that the dictators' RT negatively correlated with the difficulty of the trial, i.e., the subjective value difference between the 2 options. In other words, difficult decisions tend to take more time in the social decision domain as well.

Second, we found that observers were able to learn the dictators' preference in all conditions where they had relevant information, even when they could only observe the dictators' RT. Interestingly, compared to RT only, participants learned faster and better when they could only observe the dictators' choices or when they could observe both choices and RT, but their accuracy did not differ between the last 2 conditions. These results suggest that, in our task, participants used the RT information when no other piece of information was available, but they seemed to disregard RT when the choice information was available. We cannot rule out that there might have been an aleatoric uncertainty effect [54,55], already achieved in the choice only condition, meaning that natural constraints (such as some noise in the dictator's responses, or some sort of representational noise on the observer's end), prevented the addition of RT information on top of choice to improve participants' performance beyond this limit. In any case, since this latter result is not in line with recent literature, which suggests that people sometimes use RT on top of choice-only information to improve their inferences and predictions [24,31,37], further research is needed to dig deeper into these mechanisms. For example, contrasting choice and RT as conflicting pieces of information would be more informative to answer this specific question, which was not the main goal of the current study.

Third, we found that participants were able to predict the dictators' future decisions after having learned their preferences reaching a prediction accuracy that was higher than chance level. However, the arguably most interesting finding with respect to these predictions was that participants' RT patterns when predicting someone else's decisions matched the other person's more than their own (Fig 3C versus Fig 3D). This result strongly suggests that people are able to put themselves into someone else's shoes when predicting their decisions.

Another interesting finding is that participants showed improvement in their social preference estimation when no information (neither choice nor RT) was displayed, apart from the 2 options available to the other person. We believe that this unanticipated behavioral pattern might reflect a form of higher-order inference, where participants were able to extract information from observing the given options alone. Therefore, when no choice or RT information was given, we assume that the participant believes that the other person was asked to make very difficult decisions (and thus decisions that would be diagnostic of their social preference). Although we implemented this idea of higher-order inference in our specification of the RL model for the "none" condition and obtained support for it in our modeling results (Fig 4A), future research should investigate this further.

Over the past decades, many cognitive neuroscience studies in the field of learning and decision-making have used computational modeling to shed light on how people learn and make decisions in social contexts. Current theories suggest that 3 strategies are at play in this process [56]: vicarious RL, action imitation, and inference about others' beliefs and intentions (see [57] for a review). Of note, this distinction has been extensively discussed in developmental and comparative psychology—also referred to as "imitation versus emulation" distinction (see [1] for a review). In opposition to vicarious RL where observers learn from others' experienced outcomes, or from action imitation where observers learn from others' actions, our task involves a more complex inference process about someone else's hidden preferences. This framework usually assumes that observers update their beliefs about others' goals and intentions in a Bayesian manner [24,43,44,47,58,59], combining their prior beliefs with evidence they get from observing others' actions, both choices [45,46,57] and RTs [24,31,37]. To gain mechanistic insights into these observational learning processes, we compared such a Bayesian inference model against an RL model that takes both choices and RT into account to infer the

dictator's social preference. Instead of learning the value of options or actions, as in more conventional learning scenarios, the RL model seeks to learn the social preferences of others—in our case, the preferred allocation of money in the Dictator Game. When only choices are available, this allocation is updated in accordance with the choice (selfish versus prosocial). When only RTs are available, the updating rule depends on the speed of the decision. In case of slow decisions, the midpoint of the 2 options is used for updating. In the case of fast decisions, the observed agent is assumed to have chosen the higher-valued option, which strengthens any existing belief about the agent's preferred allocation. In our view, this implementation offers a cognitively plausible approximation that allows inferring social preferences through repeated observations of choices or RT. Accordingly, our model closely captured the performance and learning curves of observers in all the different conditions.

When comparing the RL model to a Bayesian inference model adapted to learn social preference by updating the posterior distribution, qualitative model comparison suggests that, while our participants' learning is close to optimal when they can observe choices, they substantially deviate from optimality when they can only observe RT. A potential reason why humans fall short of learning from RT in a Bayes-optimal way is its high computational complexity. The complete Bayesian solution requires one to possess an accurate generative model of the decisions and decision speed, such as a drift-diffusion model (DDM) that takes the preferred allocation, as well as the choice options into account to inform the drift rate. Furthermore, the belief distributions of a total of 5 parameters from this generative model must be updated after each observation in an accurate manner. It is conceivable that humans simplify the learning process (akin to our proposed RL model) to reduce the computational complexity and avoid getting lost in a curse of (parameter) dimensionality. A second potential reason for suboptimal performance in the RT only condition is the need to perceive the decision speed accurately for classifying an observed decision as being either fast or slow. Making incorrect classification or being uncertain in this regard will slow down learning substantially as it is likely to produce a substantial number of erroneous inferences.

Taken together, our work deviates from previous literature by challenging the expectation that, from a Bayesian perspective, people should be able to learn equally well from choices and RTs. While our empirical results are in line with the Bayesian prediction on a qualitative level, they diverge substantially from it on a quantitative level.

Our present work builds on a growing literature suggesting that RT alone should be sufficient to produce an accurate estimation of someone's preference [13,15,29]. Konovalov and Krajbich recently used a DDM without the choice data to estimate individual preferences using subjective value functions in 3 different settings: risky choice, intertemporal choice, and social preferences. Our study replicates their findings, as we were able to accurately estimate the DDM-based preference parameter from RT alone in the first sample of participants. We then took this idea a step further and showed that a second sample of participants were able to provide an accurate estimation of others' social preference when they observed their RT alone. To the best of our knowledge, this is the first time that this has been empirically tested and validated. Notably, other studies have attempted to increase out-of-sample predictive power with other indices of information processing, such as eye movements [18,60–63] or computer mouse movements [64–68]. In the neuroimaging literature, attempts have been made to move beyond brain–behavior correlations and to predict behavior from brain activity without choices [69–76] (see [77] for a review). Nevertheless, unlike eye movements or neural data, RT are easy to collect from the experimenter's point of view, and have the benefit of being directly accessible to the actual observer, making them a stronger candidate than many of the other variables mentioned above. Altogether, these and our findings point toward the richness of process data in helping to better understand and predict behavior.

An open question for future research is to elucidate the neural mechanisms that underlie the remarkable ability to learn from observing decision speed and to use this information for making predictions. Historically, brain activity tracking social inference computations was found in regions that are known to be part of the Theory of Mind network, such as dorsomedial prefrontal cortex, temporoparietal junction, and posterior superior temporal sulcus [6,45,78–80]. Nonetheless, as stated above, taking decision speed into account requires an accurate estimation of time passage, suggesting that brain regions related to time perception, such as the pre-supplementary motor area and the intraparietal sulcus [81], should play a critical role. Furthermore, our modeling indicates that a prediction error signal, which quantifies the degree of mismatch (i.e., surprise) between the anticipated and observed decision speed, should play a critical role in the RT-based updating process. Interestingly, a recent EEG study has identified such a surprise signal when participants categorized stimulus durations as being either fast or slow, and modeled this EEG signal as reflecting the distance of a diffusion particle from the anticipated threshold in a DDM-like model [82]. It is tempting to speculate that people compare the observed decision speed with their own expectations in a similar way and that the ensuing (neural) surprise signal drives the social observational learning process. Future research will need to test these predictions to further promote our understanding of how people make sense of other people's behavior.

To conclude, by investigating the relationship between RT and social preferences in the Dictator Game, we aim to contribute to the existing literature on decision-making, social cognition, and economic behavior. Our findings shed light on the intricate interplay between RT, learning and social preferences, expanding our understanding of the mechanisms underlying human decision-making in social contexts.

## Materials and methods

### Ethics statement

The research was approved by the Ethics Committee of the Faculty of Psychology and Human Movement Sciences of the University of Hamburg (approval number 2022_019) and carried out following the principles and guidelines for experiments including human participants provided in the Declaration of Helsinki (1964, revised in 2013).

### Preregistration

Our recruitment methods, task design, and procedures were preregistered on the Open Science Framework (https://osf.io/tz4dq) prior to the completion of data collection. The preregistration protocol included a within-subjects design with 2 factors (provided information: RTs and choices) and 2 levels (with/without; **Fig 1B**). A power analysis computed in G*power [83] revealed that, in order to identify an effect with the size of 0.2 (small-to-medium Cohen's f, ANOVA with repeated measures, within factors, 1 group, 4 measurements) with the power of 0.9, 46 participants should be recruited for this experiment, which uses a pure within-subject design. Preregistered hypotheses included (1) the average accuracy to be higher than chance in 3 out of 4 conditions; (2) the accuracy to increase with the amount of information provided to the participants; and (3) the accuracy to be positively correlated with their performance in the time perception task, especially in conditions where response times are displayed. For clarity of focus in our report, preregistered tests involving time perception have been described separately in **S4 Text**.

### Participants

**Dictator Game experiment.** We recruited 16 participants from a student population at the University of Basel, Switzerland, via 2 internal participant recruitment platforms (one for

psychology students who received course credits for participation, one for students of any field who received a monetary show-up fee of CHF 20 per hour). All participants gave written informed consent, and the study was approved by the local ethics committee (Ethikkommission Nordwest und Zentralschweiz).

**Main experiment.** We recruited 46 participants (38 females, 8 males, 0 N/A, aged 21.89 ± 4.05 years old) from the pool of psychology students at the University of Hamburg via an internal participant recruitment platform. Five additional participants were excluded because they did not understand the task (e.g., they did not change the preference estimation over the whole task). The Ethics Committee of the Faculty of Psychology and Human Movement Sciences of the University of Hamburg approved the study and participants provided written informed consent prior to their inclusion. To sustain motivation throughout the experiment, participants were given a monetary bonus, whose value was determined by randomly selected choices that the participants made throughout the different phases of the task.

### Behavioral tasks

**Dictator Game experiment.** After reading and signing the consent form, participants received written instructions explaining how the task worked and that their final payoff would be affected by their choices in the task. The instructions were then followed by a short training session of 4 trials, aiming at familiarizing the participants with the response modalities. In our task, point allocations were indicated by colored circles divided into a blue and a red segment. Participants were informed that the blue (resp. red) segment represented their own points (ranging from 100% to 0% of the circle) and the red (resp. blue) segment represented the points allocated to another anonymous person (ranging from 0% to 100% of the circle). Blue and red segments summed to 100%, and the color allocated to self/other was counterbalanced across participants (Fig 1). On each trial, 2 cues were presented on different sides (left /right) on the screen. The position of a given cue was randomized, such that a given cue was presented an equal number of times on the left and on the right. The points allocations were determined by increasing the "self" proportion in 10% steps. The 11 generated allocations were presented in all possible binary combinations (55 in total, not including pairs formed by the same allocation). Each pair of cues was presented 3 times, leading to a total of 165 trials. On each trial, a small noise was added to each allocation (drawn from a truncated normal distribution with fixed mean $\mu = 0$ and variance $\sigma = 0.02$, bounded between $-0.05$ and $0.05$). Participants were required to select between the allocations by pressing one of 2 keys on a standard computer keyboard. The choice window was self-paced. After the key press, the cues disappeared and were replaced by an inter-trial fixation screen, whose duration randomly varied from 1 to 4 s. At the end of the experiment, a trial was randomly selected and participants received the "self" points corresponding to the allocation they chose on this trial. In addition, the "other" points from this trial were given to another participant of the same experiment. Hence, participants received 2 bonuses: the proportion of points they chose for themselves converted to money, and the ones they received from another participant.

**Main experiment.** The main experiment was divided into 2 parts: playing the Dictator Game and observing the Dictator Game. After reading and signing the consent form, participants received written instructions explaining how the task worked and that their final payoff would be affected by their choices in the task. The first part of the task was similar to the one described in the previous paragraph, with the exceptions that (1) the combinations of allocations were presented only once, leading to a total of 55 trials; and (2) only the bonus corresponding to their own points was paid out at the end of the experiment. In the second part of

 

the task, participants were instructed that they would observe the decisions of other people who had performed the same (Dictator Game) task. After an instruction screen specifying which condition they were in (i.e., whether they would observe both choices and RT, only choices, only RT, or none), participants observed a total of 12 trials for each of the 16 dictators, in blocks of 4 dictators per condition. The order of the trials was not randomized within each dictator, but the order of the conditions was pseudo-randomized across participants, and the order of the dictators was pseudo-randomized so that each condition would display the widest range of social preferences that could be learned. For each dictator, participants were asked to estimate their social preference before observing anything, and after 4, 8, and 12 trials, leading to a total of 4 estimations per dictator. To do so, they were asked to move a tick on a slider, which simultaneously changed the visual proportion of an allocation displayed next to the slider (**S8 Fig**). After observing the 12 trials, participants were asked to indicate which allocation the dictator would have chosen in 4 previously unseen binary decision problems, presented as in the first phase of the experiment. At the end of the experiment, 1 estimation trial was randomly chosen, and an additional bonus was given to the participant, whose amount was proportional to their accuracy on this trial.

All experiments were programmed in Python using PsychoPy (www.psychopy.org).

## Trial selection

In order to maximize the likelihood of observers learning the dictators' social preferences trial-by-trial, we carefully selected which of the 165 trials would be displayed to the participants. To this end, we considered the trials which would be most informative, in terms of both choices and RT. We ran a linear regression on the RT with the subjective value distance as the independent variable (see **Results** section; **S1A Fig**):

$$RT = b_0 + b_1 * |s(\text{left}) - s(\text{right})| \tag{1}$$

We categorized the trials into slow trials ($RT > b_0$, the preference is at the midpoint between both available allocations), fast trials ($RT < b_0 + b_1 * (\text{left} - \text{right})^2$, the preference is either between 0 and the most prosocial allocation, or between the most selfish allocation and 1), and uninformative trials (all the remaining trials; **S1B Fig**). Among the informative trials, we excluded all noisy trials where dictators made an inconsistent choice, i.e., choosing the allocation with the highest distance to their social preference. For each dictator in the estimation phase, we selected the 6 fastest trials and the 6 slowest trials whose options' midpoint was the closest to the true preference (**Fig 3B**). To ensure that the trial order would not impair learning, we fitted the social preference in all possible order combinations of all possible blocks of 4 trials each. We selected the trial order which resulted in the smallest difference between the fitted and the actual preference, over all the conditions. Hence, the 12 trials of each dictator were presented in the same order for all participants. For the prediction phase, we selected the next 2 fastest trials and 2 slow "midpoint-optimizing" trials, which were presented in a random order for all participants.

## Empirical chance level

We derived the empirical chance level in the estimation trials, i.e., the average accuracy one would reach if they randomly guessed the preference for each dictator. To do so, we drew 10,000 samples from a uniform distribution of possible allocations (ranging from 0 to 1) and calculated the corresponding accuracy; the accuracy averaged over samples represents the chance level for each dictator. The chance level averaged over dictators represents the empirical chance level (0.65).

## Behavioral analyses

In the Dictator Game, we were interested in 2 variables reflecting the dictator's/participant's social preference: (1) the proportion of choices towards the more selfish allocation; and (2) the RT. In the estimation phase, we were interested in the accuracy of participant's responses, i.e., the distance between the estimated preference and the dictator's actual preference. In the prediction phase, we were interested in 3 different variables reflecting participants' strategy: (1) the accuracy, i.e., whether they selected the same allocation as the dictator did; (2) the (internal) consistency, i.e., whether they chose the allocation with the lowest distance to the last preference estimation; and (3) the RT.

For choice analyses, statistical effects were assessed using repeated measures analyses of variance (ANOVAs) with choice visibility (displayed or not) and RT visibility (displayed or not) as within-participant factors. Post hoc tests were performed using one-sample $t$ tests. We report the $t$ statistic, $p$-value (Bonferroni-corrected when applicable), and Cohen's $d$ to estimate effect size. Given the large sample size ($n = 46$), the central limit theorem allows us to assume normal distribution of our overall performance data and apply properties of normal distribution in our statistical analyses, as well as sphericity hypotheses. Regarding the comparison of correlations from dependent samples, we report Fisher's $z$ test: $z$ statistic and $p$-value. Regarding ANOVA analyses, we report Levene's test for homogeneity of variance, the uncorrected statistical, as well as Huynh–Feldt correction for repeated measures ANOVA (when applicable), $F$ statistic, $p$-value, partial eta-squared $\eta_p^2$, and generalized eta-squared $\eta^2$ (when Huynh–Feldt correction is applied) to estimate effect size.

For RT analyses, to avoid statistical fallacies arising from the assumption of normal distribution and homoskedasticity for skewed datasets, we ran GLMMs on the winsorized RT (0.05th percentile), with a Gamma distribution of the response variable and an Identity link function, with duration (duration: fast or slow), choice visibility (infoCh: displayed or not; only when predicting other), and RT information (infoRT: displayed or not; only when predicting other) as within-participant factors [84]:

$$\text{RT} \sim \text{duration*infoCh*infoRT} + (1 + \text{duration} + \text{infoCh} + \text{infoRT} | \text{observers})$$

To analyze the trials where participants chose for themselves, we only included the duration factor, as choice and RT information of another person are not shown in this phase, and adding them to the GLMM did not significantly improve the fit ($\chi^2$ (13) = 20.34, $p = 0.087$, **S1 Table**):

$$\text{RT} \sim \text{duration} + (1 + \text{duration} | \text{observers})$$

We report the estimates, standard error, $t$ statistic, and $p$-value. Post hoc tests were performed using Wilcoxon signed rank tests, for which we report the $Z$ statistic and $p$-value.

All statistical analyses were performed using MATLAB (www.mathworks.com) and R (www.r-project.org).

## Computational models

**Estimating social preference based on all data.** For the Dictator Game task, we calculated the subjective values $s(\cdot)$ for each option (left and right) at each trial $t$ using the social preference, which was estimated as a free parameter:

$$s(\text{left}_t) = 1 - (\text{left}_t - \text{P})^2$$

$$s(\text{right}_t) = 1 - (\text{right}_t - \text{P})^2, \tag{2}$$

where left (resp. right) is the objective value of the left (resp. right) option (i.e., the number of points allocated to "self") and P is the estimated social preference (free parameter that is subject-specific).

**Drift diffusion model.** To estimate social preferences in the "RT only" condition, we used a DDM, where we assumed that the drift rate in every trial is a linear function of the difference in the subjective values of the 2 options. Intuitively, individual social preferences can be identified due to the fact that longer RT should be reflective of lower drift rates and thus smaller subjective-value differences [24]. Thus, we use DDM-based probability densities to estimate the preference parameter for each dictator, given the empirical distribution of RT. Because the decision is unknown, we maximize the RT likelihood function across both choice boundaries [13]:

$$ll_{\mathrm{RT}} = \sum_t \log(f(\mathrm{RT}_t, \mathrm{choice}_t = \mathrm{left}|b, \tau, v_t)) + \log(f(\mathrm{RT}_t, \mathrm{choice}_t = \mathrm{right}|b, \tau, v_t)), \quad (3)$$

where $f$ is the response time density function, $\mathrm{RT}_t$ is the response time on a specific trial $t$, $\mathrm{choice}_t$ is the choice the dictator could have made on specific trial $t$, $b$ is the DDM decision boundary, $\tau$ is the non-decision time, and $v_t$ is the drift rate on specific trial $t$, which depends on the difference in subjective values.

**Choice-based softmax method.** For the "choice only" condition, we estimate each dictator's social preference with a softmax rule, where the probability of choosing the left option at trial $t$ is a logistic function:

$$\mathrm{Prob}_t(\mathrm{choose}_{\mathrm{left}}) = \frac{1}{1 + e^{\beta_d * (s(\mathrm{right}_t) - s(\mathrm{left}_t))}}, \quad (4)$$

where $\beta_d > 0$ is the inverse temperature parameter for the dictator $d$. High temperatures ($\beta_d \to 0$) cause the action to be all (nearly) equiprobable. Low temperatures ($\beta_d \to +\infty$) cause a greater difference in selection probability for actions that differ in their value estimates [52]. The social preference and the temperature are free parameters that can be estimated for each dictator individually by maximizing a likelihood function [13]:

$$ll_{\mathrm{Ch}} = \sum_t \log(\mathrm{Prob}_t(\mathrm{choose}_{\mathrm{left}})) \cdot 1(\mathrm{choice}_t = \mathrm{left}) + \log(1 - \mathrm{Prob}_t(\mathrm{choose}_{\mathrm{left}})) \cdot 1(\mathrm{choice}_t$$
$$= \mathrm{right}), \quad (5)$$

where at each trial $t$, $\mathrm{choice}_t$ is the choice made by the dictator and $1(\cdot)$ is the indicator function. Notably, the close correspondence of the softmax (or logit) choice model and the DDM has been elaborated in previous work (e.g., [85]).

**Learning social preference based on sequential observations.** The goal of our learning models is to infer the observed dictator's social preference over trials and to choose the best (i.e., subjective-value maximizing) allocation in the prediction phase. We compared 2 alternative computational models: an adapted reinforcement learning model which updates the subjective value of the social preference with a delta-rule and a Bayes-optimal model that integrates the posterior likelihood over a set of predefined parameter prior distributions.

**Reinforcement learning models.** To model participants' behavior, we designed 2 modified versions of the standard RL model [52]. In both models, the initial estimated preference $P_0$ was included as a free parameter. At each trial $t$, the estimated preference $P$ is updated with a delta rule [51]:

$$P_t = P_{t-1} + \alpha * \delta_t, \quad (6)$$

where $\alpha$ is the learning rate and $\delta_t$ is a prediction error term, calculated as the difference

between the outcome $O_t$ (defined below) and the current estimation:

$$\delta_t = O_t - P_{t-1}. \tag{7}$$

The outcome $O_t$ depends on the amount of information provided to the participant.

When both choices and RT were displayed, the trial was categorized as either fast or slow, depending on whether the trial RT was longer than the average RT observed for this dictator. If the trial was slow, the outcome was computed as the midpoint between the objective values of both options (proportion of points for "self"), reflecting the intuition that both options were likely equidistant from the preferred allocation. If the trial was fast, the outcome was computed as 1 or 0, depending on whether the chosen allocation was the more selfish or the more prosocial one:

$$O_t = \begin{cases} \dfrac{\text{left}_t + \text{right}_t}{2} \text{ if } RT_t > \text{mean}(RT_{1:t}) \\ 1 \text{ if choice}_t = \max\{\text{left}_t, \text{right}_t\} \\ 0 \text{ if choice}_t \neq \max\{\text{left}_t, \text{right}_t\} \end{cases} \tag{8}$$

When only the choices were displayed, the outcome was computed as 1 or 0, depending on whether the chosen allocation was the more selfish or the more prosocial one:

$$O_t = \begin{cases} 1 \text{ if choice}_t = \max\{\text{left}_t, \text{right}_t\} \\ 0 \text{ if choice}_t \neq \max\{\text{left}_t, \text{right}_t\} \end{cases} \tag{9}$$

When only RT were displayed, the trial was categorized as either fast or slow as specified above. If the trial was slow, the outcome was computed as the midpoint between the objective values of both options (proportion of points for "self"). If the trial was fast, the outcome was computed as 1 or 0, assuming that the allocation with the highest subjective value $s(\cdot)$, as given in **Eq 2**, was chosen:

$$O_t = \begin{cases} \dfrac{\text{left}_t + \text{right}_t}{2} \text{ if } RT_t > \text{mean}(RT_{1:t}) \\ 1 \text{ if } \max\{s(\text{left}_t), s(\text{right}_t)\} = \max\{\text{left}_t, \text{right}_t\} \\ 0 \text{ if } \max\{s(\text{left}_t), s(\text{right}_t)\} \neq \max\{\text{left}_t, \text{right}_t\} \end{cases} \tag{10}$$

Intuitively, this implies that in the case of observing a fast decision, the observer is assumed to believe that the option with the higher subjective value was chosen and to update the social preference accordingly. When no information was displayed, the outcome was computed as the midpoint between the objective values of both options (proportion of points for "self") at each trial:

$$O_t = \frac{\text{left}_t + \text{right}_t}{2} \tag{11}$$

Intuitively, this implies that in case of receiving no information, the observer is assumed to believe that the dictator was asked to make very difficult decisions (and thus decisions that would be diagnostic of their social preference). In the version of the RL model presented in the main text, the outcome was computed as a weighted sum between the choice-related

information and the RT-related information, with the addition of a weight parameter $0<\omega<1$:

$$O_t = O_{\mathrm{Ch},t}*(1-\omega) + O_{\mathrm{RT},t}*\omega \qquad (12)$$

To fit the model to the data, maximum likelihood estimation was applied by minimizing the deviance between the data and the model. For estimation trials, the negative log-likelihood was computed as follows:

$$ll_t = -\log(\phi(D_t, P_t, \sigma)) \qquad (13)$$

where $\phi(D_t, P_t, \sigma)$ represents the value of a normal distribution at $D_t$ (actual participant's estimation) with mean $P_t$ (model estimation) and standard deviation $\sigma$ (fitted as a free parameter). For the prediction trials, the negative log-likelihood was computed as follows:

$$ll_t = -\frac{1}{1 + e^{\beta_p * (s(\mathrm{unchosen}_t) - s(\mathrm{chosen}_t))}}, \qquad (14)$$

where $\beta_p > 0$ is the inverse temperature parameter for the participant $p$ and $s(\cdot)$ represents the subjective value of an allocation, computed as in **Eq 2**. To avoid local minima, model fitting was performed with 50 different initial parameter values, randomly drawn from prior distributions, which we took to be Beta(1.1,1.1) for the learning rate $\alpha$, Gamma(1.2,5) for the inverse temperature $\beta_p$, and a fixed number for the standard deviation $\sigma$ [86].

We modeled participants' choice behavior using a softmax decision rule representing the probability of a participant choosing the left allocation:

$$\mathrm{Prob}_t(\mathrm{choose}_{\mathrm{left}}) = \frac{1}{1 + e^{\beta_p * (s(\mathrm{right}_t) - s(\mathrm{left}_t))}}, \qquad (15)$$

where $\beta_p > 0$ is the inverse temperature parameter for the participant $p$. High temperatures ($\beta_p \to 0$) cause the action to be all (nearly) equiprobable. Low temperatures ($\beta_p \to +\infty$) cause a greater difference in selection probability for actions that differ in their value estimates [52].

**Bayes-optimal model.**   Based on prior work suggesting that observers might infer other people's choice processes via Bayesian inferences [24,43–47], we tested a model that assumes observers estimate the dictator's preference using such a Bayesian framework. More specifically, this framework suggests observers infer the optimal parameters by maximizing the posterior distribution of the parameter set, given all the evidence collected so far. In other words, we designed a benchmark Bayes-optimal model which assumes that observers seek to know what is the most likely set of parameters (of a model they assume to be the generative model of the decisions) for this dictator given the observation. To make such an inference, observers must have a generative model of the dictator's decision-making process (or a model that can be expected to come reasonably close to the true generative model, which itself is unknown). Following previous work [24], we assumed this model to be a DDM, which indeed provides an excellent account of both choices and RT in our Dictator Game task (**S9 Fig**). In this framework, the drift rate depends on the social preferences and choice options as specified above. On each trial, we computed the likelihood of the observed behavior given a set of parameters from the joint parameters space as follows.

When both choices and RT were displayed, the likelihood was computed as the probability density function of the Wiener first-passage time (WFPT) distribution [87], i.e., the diffusion process given the observed choice and RT (**Fig 1D, rightmost panel**). When only choices were displayed, the likelihood was computed as a softmax function (**Eq 14 and Fig 1D, middle panel**). When only RT were displayed, the likelihood was computed as the probability density function of the WFTP distribution given the RT, and integrated over both possible choices

(Fig 1D, **leftmost panel**). When no information was displayed, the likelihood did not differ from the priors for the joint parameter space, which we took to be Beta(3.5,3) for the estimated preference $0<P<1$, Gamma(1.2,5) for the temperature $0<\beta<100$, Gamma(2,2) for the boundary separation $0.1<\alpha<10.1$, Normal(0,5) for the drift rate $0<v<20$, and a uniform distribution for the non-decision time $0.1<T_{er}<0.5$. The posterior distribution was then computed and used as a prior for the next trial.

## Supporting information

**S1 Text. Trial selection prior to running the main experiment.**
(DOCX)

**S2 Text. Observers own preference impact their uniformed guesses.**
(DOCX)

**S3 Text. Qualitative model comparison favors the RL model.**
(DOCX)

**S4 Text. Time perception and Social Value Orientation score.**
(DOCX)

**S5 Text. Prediction phase's GLMM sanity check.**
(DOCX)

**S1 Fig. Trial selection procedure and corresponding RT. (A)** Illustration of the single-peaked model regression with 2 choice options A and S, with $A>S$ without loss of generality. **(B)** Proportions of all 165 trials performed by the dictators in the Dictator Game experiment, categorized using the single-peaked model. **(C)** Dictators' average RT in the selected 6 fast trials ("short RT") and 6 slow trials ("long RT"), as seen by the observers in each of the conditions. Points indicate individual average, shaded areas indicate probability density function, 95% confidence interval, and SEM. $N = 46$. Data and analysis scripts underlying this figure are available at https://github.com/sophiebavard/beyond-choices.
(PDF)

**S2 Fig. Observers' estimations as a function of their own preference. (A)** Observers' average first estimation as a function of their own social preference. $N = 46$. **(B)** Observers' average estimation per condition as a function of their own social preference. ρ: Spearman's coefficient. $N = 48$. In all panels, ns: $p > 0.05$, $*p < 0.05$, $**p < 0.01$, $***p < 0.001$. Data and analysis scripts underlying this figure are available at https://github.com/sophiebavard/beyond-choices.
(PDF)

**S3 Fig. Additional behavioral results in the estimation phase.** Reported fourth and last estimation per dictator, averaged over observers, as a function of the true preference of each dictator, for each condition. ρ: Spearman's coefficient. $N = 16$. In all panels, ns: $p > 0.05$, $**p < 0.01$, $***p < 0.001$. Data and analysis scripts underlying this figure are available at https://github.com/sophiebavard/beyond-choices.
(PDF)

**S4 Fig. RL model predictions. (A)** Estimated difficulty extracted from RL model predictions as a function of the estimated difficulty from behavioral data from observers and dictators, after the estimation phase, for trials from the prediction phase. Each point represents one average trial difficulty for each duration (fast/slow) for each observer, averaged over dictators and conditions. $N = 92$. **(B)** RL model predictions for the proportion of choices towards the left

option in the prediction phase, as a function of the behavioral data. **(C)** RL model predictions for the fourth and last estimation per observer per observed dictator, as a function of the reported fourth and last estimation, for each condition. $N = 184$. ρ: Spearman's coefficient. In all panels, ***$p < 0.001$. Data and analysis scripts underlying this figure are available at https://github.com/sophiebavard/beyond-choices.
(PDF)

**S5 Fig. Additional qualitative model comparison.** Average accuracy for the last estimation predicted by the RL model **(A)** and BO model **(B)** for each condition, averaged over trials and dictators, as a function of the observers' behavioral accuracy. In all panels, $N = 46$, ns: $p > 0.05$, *$p < 0.05$, **$p < 0.01$, ***$p < 0.001$. Data and analysis scripts underlying this figure are available at https://github.com/sophiebavard/beyond-choices.
(PDF)

**S6 Fig. Qualitative model comparison for full model space.** Top: Simulated data (colored dots) superimposed on behavioral data (colored curves) representing the accuracy in the estimation phase for the main RL model **(A)**, the basic RL model **(B)**, the BO model with informative priors **(C)**, and the BO model with uninformative uniform priors **(D)** in each condition. Shaded areas represent SEM. $N = 46$. Bottom: accuracy predictions the main RL model **(A)**, the basic RL model **(B)**, the BO model with informative priors **(C)**, and the BO model with uninformative uniform priors **(D)** as a function a behavioral accuracy in the estimation phase for the last estimation of each participant in each condition. Dashed line represents identity. $N = 184$. Data and analysis scripts underlying this figure are available at https://github.com/sophiebavard/beyond-choices.
(PDF)

**S7 Fig. Between-group comparisons in the estimation phase.** Subset of observers' accuracy for each estimation as a function of the condition (choice and RT visibility). **(A)** Observers whose first condition was "none." **(B)** Observers whose first condition was "RT." **(C)** Observers whose first condition was "Ch." **(D)** Observers whose first condition was "both." **(E)** Observers whose last condition was "none." **(F)** Observers whose last condition was "RT." **(G)** Observers whose last condition was "Ch." **(H)** Observers whose last condition was "both." Data and analysis scripts underlying this figure are available at https://github.com/sophiebavard/beyond-choices.
(PDF)

**S8 Fig. Visual representation of the estimation phase.** The figure represents the screen seen by observers to indicate what they thought the dictator's preference was, by dragging-and-dropping a red (resp. blue for counterbalanced observers) tick on a slider. The < Continue with space bar > line was only displayed after they had made one first click, to avoid perseveration effects. Translated from German for illustration purposes. Data and analysis scripts underlying this figure are available at https://github.com/sophiebavard/beyond-choices.
(PDF)

**S9 Fig. DDM simulations on the Dictator Game task.** The DDM model was fitted on the Dictator Game data for the 16 dictators, here represented in an increasing order based on their social preference (estimated from their behavioral choices). The DDM is able to match dictators' behavior both in terms of choices (top panels) and RT (bottom panels). Data and analysis scripts underlying this figure are available at https://github.com/sophiebavard/beyond-choices.
(PDF)

**S10 Fig. Results of additional experiments. (A)** Observers' accuracy in the time perception task as a function of the difficulty of the trial, i.e., the time interval between 2 stimuli. Points indicate individual average, shaded areas indicate probability density function, 95% confidence interval, and SEM. $N$ = 46. **(B)** Observers' post-task SVO score as a function of their pre-task SVO score and their social preference extracted from their choices in the Dictator Game (DG). Black dashed lines represent categorical boundaries: competitiveness/individualism/prosociality/altruism. Red dashed lines represent a change of category from pre- to post-task scores. SVO: Social Value Orientation scale; DG: Dictator Game; $N$ = 46. Data and analysis scripts underlying this figure are available at https://github.com/sophiebavard/beyond-choices. (PDF)

**S1 Table. Results from GLMM fitted on observers' RT in the prediction phase.** The GLMM (generalized linear mixed model with Gamma distribution and identity link function) was fitted on the observers' RT, with choice visibility in the estimation phase, RT visibility in the estimation phase, and trial duration (i.e., whether the dictator's RT was short or long), as independent variables. Denotation: Du = duration (fast or slow), Ch = choice visibility (displayed or not), RT = RT visibility (displayed or not), ***$p < 0.001$. Data and analysis scripts underlying this figure are available at https://github.com/sophiebavard/beyond-choices. (DOCX)

## Acknowledgments

We thank Marie Habermann, Julia Hecht, and Anne Kaufmann for their help in data collection.

## Author Contributions

**Conceptualization:** Sophie Bavard, Erik Stuchlý, Arkady Konovalov, Sebastian Gluth.

**Data curation:** Sophie Bavard.

**Formal analysis:** Sophie Bavard.

**Funding acquisition:** Sebastian Gluth.

**Investigation:** Sophie Bavard, Sebastian Gluth.

**Methodology:** Sophie Bavard, Erik Stuchlý, Arkady Konovalov, Sebastian Gluth.

**Project administration:** Sophie Bavard, Sebastian Gluth.

**Supervision:** Sebastian Gluth.

**Validation:** Sophie Bavard, Erik Stuchlý, Arkady Konovalov, Sebastian Gluth.

**Visualization:** Sophie Bavard.

**Writing – original draft:** Sophie Bavard, Erik Stuchlý, Arkady Konovalov, Sebastian Gluth.

**Writing – review & editing:** Sophie Bavard, Erik Stuchlý, Arkady Konovalov, Sebastian Gluth.

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
