## [Editor Report · Decision Letter 0]

5 Dec 2023

Dear Dr Bavard, 

Thank you for submitting your manuscript entitled "Beyond choices: humans can infer social preferences from decision speed alone" for consideration as a Research Article by PLOS Biology.

Your manuscript has now been evaluated by the PLOS Biology editorial staff as well as by an academic editor with relevant expertise and I am writing to let you know that we would like to send your submission out for external peer review.

Once your full submission is complete, your paper will undergo a series of checks in preparation for peer review. After your manuscript has passed the checks it will be sent out for review. To provide the metadata for your submission, please Login to Editorial Manager (https://www.editorialmanager.com/pbiology) within two working days, i.e. by Dec 07 2023 11:59PM.

Kind regards,

Christian

Christian Schnell, PhD

Senior Editor

PLOS Biology

cschnell@plos.org

---

## [Decision Letter · Decision Letter 1]

19 Jan 2024

Dear Dr Bavard,

Thank you for your patience while your manuscript "Beyond choices: humans can infer social preferences from decision speed alone" went through peer-review at PLOS Biology. Your manuscript has now been evaluated by the PLOS Biology editors, an Academic Editor with relevant expertise, and by several independent reviewers.

In light of the reviews, which you will find at the end of this email, we are pleased to offer you the opportunity to address the comments from the reviewers in a revision that we anticipate should not take you very long. We will then assess your revised manuscript and your response to the reviewers' comments with our Academic Editor. As you will see, one of the comments refers to the inaccessibility of the data repository and Reviewer 3 indicated that they would like to review the data and code for an in-depth review. 

**IMPORTANT - SUBMITTING YOUR REVISION**

*Resubmission Checklist*

*Published Peer Review*

*PLOS Data Policy*

*Blot and Gel Data Policy*

Sincerely,

Christian

Christian Schnell, PhD

Senior Editor

PLOS Biology

cschnell@plos.org

REVIEWS:

Reviewer #1: Reviewer comments

Summary

The authors investigate the role of response times (RT) in how individuals learn about others' social preferences. The key novel contribution is the finding that RT is predictive of social choice preferences, meaning that observers could infer others' choices were more prosocial versus more selfish when provided only with information about RT (i.e., without information about choices). The authors argue that this deviates from the predictions of an optimal Bayesian model in which people are expected to learn equally well from choices and RT information. 

Comments

1. Regarding the Bayesian optimal observer model, something that was unclear in the manuscript is why or in what sense is this model optimal? Does it maximize some performance measure (e.g., accuracy or reward rate per unit time)? A few sentences clarifying this would be beneficial. 

2. From the description in the Materials and Methods section, the Bayesian optimal model seems to be comprised of several different models (e.g., Weiner first-passage time, soft-max, WFPT integrated over both choices, and the 'prior equals likelihood' model) which are applied depending on the type of data provided (RT/choices/both/none). The manuscript would benefit from giving a brief description of and motivation for this 'benchmark' model earlier on, such as when the model is first mentioned in the introduction. Doing so would help readers understand the reason for choosing these models as a benchmark for comparison. 

3. It is stated in a few places (Abstract/Intro/Discussion) that "from a Bayesian perspective, people should be able to learn equally well from choices and RTs". I think this statement needs a bit more motivation/unpacking as to why choices and RTs should be equally informative when predicting choices. I would think that choice information would be more informative (weighted more heavily when updating a prior) when the task is to infer people's choice preferences. For example, if my task is to infer someone's choice preferences, and I'm given the option to either view their 10 previous choices or 10 previous RTs, I can't really think of a situation in which I'd choose the RT information. If, on the other hand, my task was to infer someone's mean RT, then the RT information would indeed be more informative, and I would disregard the choice information. The reported behavioural results appear to support this strategy: "These results suggest that, in our task, participants used the RT information when no other piece of information was available, but they seemed to disregard RT when the choice information was available" (p.12). Thus, the Bayesian optimal model seems to be somewhat of a 'straw man' rather than a serious competitor against the other models presented in the paper. This could be remedied by providing some further explanation and motivation for this model.

4. Aside from these minor points, I thought the paper was well written and provided a comprehensive analysis of an interesting dataset. 

Reviewer #2: The authors examine whether or not people use different combinations of the speed and choices of other peoples decisions when making inferences about social preferences of others. Their main conclusions are the people are able to learn other peoples' social preferences even when they only observed the response time of others'. This is a very interesting finding and, as far as I know, the extent to which response time can be used to infer the social preferences of others has not been carefully examined, which makes this study a novel and exciting addition to the literature. 

The manuscript is short and sweet -- they use a direct and straightforward design and show a clear main effect of response time in the accuracy of the inferences drawn about others' social preferences. They also use the response time to update a simple reinforcement learning model to allow the model to learn to predict subsequent choice and response times. The model is already fairly established in other applications, so simply adding response time in the current context is a natural step. The authors also use a Bayesian optimal model as a point of comparison and evaluating their RL model. Although I didn't examine this BO model closely, the authors show rather large qualitative (and quantitative) differences in the accuracy of the two models, with the RL model clearly outperforming the BO model. 

Given the straightforward approach (in terms of the experimental design and computational modeling) as well as the very clear experimental and computational modeling results, I really don't have much to offer for suggested improvements. The authors provided a very accessible manuscript and have cited all the relevant literature that I can think of, so I would recommend acceptance. 

Reviewer #3: The manuscript addresses an experiment in which participants observe information about others' decisions or response times to make decisions in the context of prosocial behavior, and finds that people can infer others' preferences from the process by which they make decisions. There is much to like about the paper, and although the details of the developed learning model are outside of my area of expertise to assess, I find myself to be enthusiastic overall. 

However, although it appears that there is documentation of the materials, data and code available on github, the link provided lead me to a 404 error message. Regarding the OSF project related to the preregistration, I believe I did not have reading rights. Therefore, it was not possible for me to review the paper in depth. I would be happy to do so in another round of reviews. I did find and review the preregistration, which clearly stipulated the manipulated conditions and hypotheses but gave only sparse indications of the overall procedure or analyses to be expected. In the manuscript, it would be great if it was even easier to spot which results related to preregistered hypotheses. In any case, I find it highly commendable that the authors provided a preregistration.

In addition, I have a few comments about the behavioral results that I believe should be addressed in a revision of the manuscript. 

Learning about Dictators' Preferences (Estimation Phase)

In the results section (p. 5, bottom paragraph), the authors report that observers' learning of the dictators' preferences was better than chance overall, and separately for the four conditions induced (RT only, choice only, both, none), in correspondence with the preregistered hypothesis that "(1) the mean accuracy of participants to predict the dictators' social preference will be higher than chance in three out of four conditions, [and] (2) this accuracy will increase with the amount of information provided to the participants". The paragraph is quite densely written, so that perhaps it might be helpful to add a bit of explanation back in. For instance: starting with line 178, do you report first the overall effect of learning (in all conditions), then the effect separately for the RT only condition, then a post-hoc contrast between the choice only and both conditions, and then the effect separately for the none condition? Perhaps a table would help summarize these results efficiently, and give you more space for elaborating in the text. 

Moreover, I expected to see an interaction effect reported about the effect of condition x time of measurement (before first trial, after first trial, after second trial, etc.) on the perception of the dictators' preferences. It seems to me like the perceptions of the dictators' preferences would probably get better across the four trials, maybe particularly so in the both condition and perhaps not at all in the none condition. 

Regarding the dictators' preference in the methods section, it didn't quite become clear to me how the participants indicated what they thought the dictators' preferences were. 

Extrapolation to Unseen Decisions

Minor point: I noticed that the terminology and the perspective on the data changed a bit in this section. Where you referred to RT only condition before, you now refer to the RT visibility (which presumably also addressed the both condition). For consistency and to increase the ease with which readers can follow the presented work, perhaps it might be good to keep the perspective on the data constant throughout, or to alert the readers to the difference?

---

## [Decision Letter · Decision Letter 2]

19 Mar 2024

Dear Dr Bavard,

Thank you for your patience while we considered your revised manuscript "Beyond choices: humans can infer social preferences from decision speed alone" for consideration as a Research Article at PLOS Biology. Your revised study has now been evaluated by the PLOS Biology editors and the original reviewers.

In light of the reviews, which you will find at the end of this email, we are pleased to offer you the opportunity to address the remaining points from the reviewers in a revision that we anticipate should not take you very long. We will then assess your revised manuscript and your response to the reviewers' comments with our Academic Editor aiming to avoid further rounds of peer-review, although might need to consult with the reviewers, depending on the nature of the revisions.

As you can see, Reviewer 3 was not able to access the data to review them. One issue is that one of the OSF repositories does not seem to contain the required data (https://osf.io/fseyw/) and that the github repository contains scripts and data as matlab files, making them inaccessible for the reviewers (and some of our readers). Would you able to provide these data and scripts in a more accessible way? Please don't hesitate to contact me if you have any questions or would like to discuss alternative options. 

**IMPORTANT - SUBMITTING YOUR REVISION**

*Resubmission Checklist*

*Published Peer Review*

*PLOS Data Policy*

*Blot and Gel Data Policy*

Sincerely,

Christian

Christian Schnell, PhD

Senior Editor

PLOS Biology

cschnell@plos.org

REVIEWS:

Reviewer #1 (Russell J. Boag): The authors have addressed my comments satisfactorily.

Reviewer #3: To the degree that the manuscript coincides with my expertise, I found most of my comments addressed appropriately. However, I continue to quarrel with details relating to the Open practices for this manuscript.

Unfortunately, the linked OSF project was empty. Therefore, performing a review based on the data and materials that should presumably be contained therein was not possible. I think it would be great if materials, such as the program used to elicit responses from participants, were made available in this repository. Given that PLOS Biology is an outlet that is committed to transparency regarding research protocols, this seems like an important step to me. 

In the github project, I could not locate a file that looked like it contained materials, too. I assume the data might be stored in .mat files (e.g., data_fig.mat), which makes them inaccessible to users like me who are not running Matlab. Might I suggest saving a more accessible version of the raw data, in line with the FAIR principle of interoperability?

---

## [Decision Letter · Decision Letter 3]

24 Apr 2024

Dear Sophie,

Thank you for your patience while we considered your revised manuscript "Beyond choices: humans can infer social preferences from decision speed alone" for publication as a Research Article at PLOS Biology. This revised version of your manuscript has been evaluated by the PLOS Biology editors, the Academic Editor and one of the original reviewers.

Based on the reviews and on our Academic Editor's assessment of your revision, we are likely to accept this manuscript for publication, provided you satisfactorily address the following data and other policy-related requests.

* We would like to suggest a different title to improve readability: "Humans can infer social preferences from decision speed alone"

* Please add the links to the funding agencies in the Financial Disclosure statement in the manuscript details.

* All research involving human participants must have been approved by the authors' Institutional Review Board (IRB) or an equivalent committee, and must have been conducted according to the principles expressed in the Declaration of Helsinki. Please provide the approval number and a statement that your study has been conducted according to the principles expressed in the Declaration of Helsinki in the manuscript.

DATA POLICY:

Regardless of the method selected, please ensure that you provide the individual numerical values that underlie the summary data displayed in the following figure panels as they are essential for readers to assess your analysis and to reproduce it: 1D, 2A, 3ABCD.

CODE POLICY

Per journal policy, if you have generated any custom code during the curse of this investigation, please make it available without restrictions upon publication. Please ensure that the code is sufficiently well documented and reusable, and that your Data Statement in the Editorial Manager submission system accurately describes where your code can be found. As the code that you have generated to obtain the data is important to support the conclusions of your manuscript (as also mentioned by Reviewer 3), its deposition is required for acceptance.

We expect to receive your revised manuscript within two weeks. 

*Published Peer Review History*

*Press*

Sincerely,

Christian

Christian Schnell, PhD

Senior Editor

cschnell@plos.org

PLOS Biology

Reviewer remarks:

Reviewer #3: Thanks for taking up my suggestions to improve the transparency of the documentation. It seems like you missed out on one request (to also share materials used to obtain the data, e.g., stimulus materials and instructions, program run to present stimuli). I hope these materials can be made available nevertheless.

---

## [Editor Report · Decision Letter 4]

3 May 2024

Dear Sophie,

Thank you for your patience while we considered your revised manuscript "Beyond choices: humans can infer social preferences from decision speed alone" for publication as a Research Article at PLOS Biology. This revised version of your manuscript has been evaluated by the PLOS Biology editors.

Thank you for addressing most of the editorial requests. However, a few items were not fully addressed:

*) Title: I understand that you'd like to keep the choices in the title. However, we do not allow split titles for Research Articles. Would you be able to to suggest a title without the :? I've tried a couple of variants but they were all worse than the original suggestion. It would be helpful if you could send me your suggestions via email to cschnell@plos.org before submitting the revision, so we can discuss this without you going through another round via Editorial Manager. 

*) Thank you for uploading the stimulation data to the github repository. However, the GitHub repository could be changed after publication. Therefore, can 

you please archive this version of your publicly available GitHub repository to Zenodo? Please update the Data Accessibility Statement afterwards accordingly (you are welcome to also provide the GitHub access information). See the process for doing this here: https://docs.github.com/en/repositories/archiving-a-github-repository/referencing-and-citing-content

*) Source data: We ask that all individual quantitative observations that underlie the data summarized in the figures and results of your paper be made available in one of the following forms:

Regardless of the method selected, please ensure that you provide the individual numerical values that underlie the summary data displayed in the following figure panels as they are essential for readers to assess your analysis and to reproduce it: 1D, 2A and 3ABCD.

You can see an example of this in this paper: https://journals.plos.org/plosbiology/article?id=10.1371/journal.pbio.3002591#pbio.3002591.s012

We expect to receive your revised manuscript within two weeks. 

*Published Peer Review History*

*Press*

Sincerely,

Christian

Christian Schnell, PhD

Senior Editor

cschnell@plos.org

PLOS Biology

---

## [Editor Report · Decision Letter 5]

21 May 2024

Dear Dr Bavard,

Thank you for the submission of your revised Research Article "Humans can infer social preferences from decision speed alone" for publication in PLOS Biology. On behalf of my colleagues and the Academic Editor, Thorsten Kahnt, I am pleased to say that we can in principle accept your manuscript for publication, provided you address any remaining formatting and reporting issues. These will be detailed in an email you should receive within 2-3 business days from our colleagues in the journal operations team; no action is required from you until then. Please note that we will not be able to formally accept your manuscript and schedule it for publication until you have completed any requested changes.

PRESS

Sincerely, 

Christian

Christian Schnell, PhD

Senior Editor

PLOS Biology

cschnell@plos.org